# Effects of a level-based immersive virtual reality physical therapy program on static and dynamic balance in Parkinson's disease: Protocol for a randomized controlled trial

Lorena Morcillo-Martínez[1], María Sanz-Mármol[1], Adriana Isabel Romanos-Navarrete[2] Sandra Calvo [2,3]*, Lilian Le Roux-Ethève[1], Laura Esteban-Repiso[1], Paula Córdova-Alegre[1], Carolina Jiménez-Sánchez [1], Aitor Garay-Sánchez[4], Natalia Brandín-de la Cruz[1]

**1** Department of Physical Therapy, Faculty of Health Sciences, Universidad San Jorge, Villanueva de Gállego, Zaragoza, Spain, **2** Department of Physiatry and Nursing, Faculty of Health Sciences, University of Zaragoza, Zaragoza, Spain, **3** Institute for Health Research Aragón (IIS Aragón), Zaragoza, Spain, **4** Miguel Servet University Hospital, Institute for Health Research Aragón (IIS Aragón), Zaragoza, Spain

* sandracalvo@unizar.es

## Abstract

### Background

Parkinson's disease (PD) is a neurodegenerative disorder characterized by postural instability, which significantly contributes to an increased risk of falls (occurring in 45–68% of persons with PD annually), leading to greater functional dependence, social isolation, and a significant decrease in quality of life. The aim of this randomized clinical trial is to evaluate the effects of an immersive virtual reality (IVR) intervention on static and dynamic balance, gait speed, functional mobility, lower limb strength, and quality of life in people with PD. Furthermore, the study will assess treatment adherence, participant satisfaction, as well as the potential occurrence of adverse effects associated with the application of IVR.

### Methods

A 10-month, single-blind, two-arm randomized controlled trial will be conducted with participants aged 55–90 years diagnosed with PD and classified between stages 2.5 and 4 on the modified Hoehn and Yahr scale. Participants will be randomly assigned to one of two groups: an experimental group and a control group. The intervention will consist of an 8-session program. The first part of each session, common to both groups, will include a standardized strength training program. In the second part, the experimental group will receive a physical therapy balance program using IVR, while the control group will follow a usual physical therapy balance exercise program.

**Data availability statement:** No datasets have been generated or analyzed yet, as this is a study protocol. Upon completion of the trial, the data will be available from the corresponding author on reasonable request, in accordance with privacy and data protection regulations.

**Funding:** This research has been funded by the Government of Aragon (grant number: B61_23D) and has received support from San Jorge University (USJ) through the 2024-2026 Open Access Program. The beneficiary of this funding is the iPhysio research group at San Jorge University, to which several authors of this publication belong.

**Competing interests:** The authors have declared that no competing interests exist.

**Abbreviations:** PD, Parkinson's Disease; IVR, Immersive Virtual Reality; CNS, Central Nervous System; VR, Virtual Reality; NIVR, Non-Immersive Virtual Reality; SIVR, Semi Immersive Virtual Reality; MMSE-MEC, Mini-mental State Examination; SSQ, Simulator Sickness Questionnaire; mFC, Modified Fitness Counts; Mini-BESTest, Mini Balance Systems Test; TUG, Time Up & Go; TUG-DT, Time Up & Go Dual Task; 10MWT, 10-Meter Walk Test; 30CST, 30-Second Sit-to-Stand Test; SFT, Senior Fitness Test; PDQ-39, Parkinson Disease Questionnaire-39; ITT, Intention-To-Treat; LOCF, Last Observation Carried Forward; BBS, Berg Balance Scale; MTG, Moving Through Glass; STS, Sit to Stand.

## Discussion

This study will examine whether IVR can improve static and dynamic balance rehabilitation in Parkinson's disease beyond usual physical therapy, addressing real-world motor and dual-task challenges. By providing enriched sensory input and adaptive tasks, the IVR seeks to optimize motor learning and functional mobility. Demonstrating safety and clinically meaningful improvements would support IVR as a scalable and engaging tool to reduce risk of falls and enhance quality of life in PD.

## Trial registration

ClinicalTrials.gov NCT07274514

## Introduction

Parkinson's disease (PD) is the second most prevalent neurodegenerative disorder. Its incidence is estimated to range from 5 to 35 cases per 100,000 people, with a marked increase observed with advancing age, rising by 5- to 10-fold between ages 60 and 90 [1]. The number of individuals diagnosed with PD worldwide is expected to reach approximately 1.93 million by 2030, with age-adjusted incidence rates projected to be 27 per 100,000 population [2]. Although the exact etiology remains unclear, PD is considered a multifactorial condition resulting from complex interactions between genetic and environmental factors. Genetic components are estimated to account for approximately 5–10% of cases [3–5]. Additionally, exposure to environmental risk factors, including toxins, pesticides, bacteria, and viruses, has been associated with an increased likelihood of developing PD, possibly due to inflammatory responses mediated by microglial activation [4].

PD is primarily characterized by the progressive degeneration of dopaminergic neurons in the substantia nigra, a critical component of the basal ganglia circuitry. This neurodegeneration leads to reduced dopamine levels, resulting in impaired motor control and disrupted communication between the basal ganglia and other regions of the central nervous system (CNS), such as the motor and premotor cortices [6]. The reduction in dopaminergic input also triggers compensatory changes in other neurotransmitter systems, particularly increased cholinergic and glutamate activity. These changes disturb the neurochemical balance within the striatal pathways, compromising the precise regulation of motor functions and contributing to the development of PD symptoms [7].

The clinical manifestations of PD include both motor and non-motor symptoms, usually appearing around 60 years of age, although early-onset cases also occur [8]. Non-motor symptoms, which often precede motor signs by up to 10 years, are attributed to the involvement of other CNS regions, including the nucleus basalis of Meynert, olfactory bulb, dorsal motor nucleus of the vagus, and peripheral autonomic ganglia and pathways [9,10]. These symptoms encompass neuropsychiatric disturbances [8,9,11]; autonomic dysfunction [9,10]; and oculomotor abnormalities [10,12]. Motor impairments typically involve resting tremor, rigidity, bradykinesia, postural

instability, and gait abnormalities [5]. These clinical manifestations are often exacerbated by increased muscular stiffness in the trunk and extremities, which further contributes to postural dysfunction and balance impairments. These issues significantly affect daily functioning [13] and lead to abnormal postural and movement patterns, resulting in deficits in both static and dynamic balance. Postural instability and gait disturbances usually develop between 10–15 years after the onset of PD and are among its most disabling manifestations. In particular, postural instability is a major source of morbidity, as it frequently leads to recurrent falls (occurring in 45–68% of patients with PD annually) [14,15] reduced functional independence, and social isolation [1]. These consequences contribute to a decline in quality of life and place a considerable burden on caregivers and healthcare systems [14].

While pharmacological treatments, such as levodopa and dopamine agonists, and surgical options like deep brain stimulation, can alleviate some symptoms [11,16,17], their efficacy tends to decrease over time, and they do not prevent disease progression [18]. Therefore, complementary therapies, including physical therapy, play a crucial role in the long-term management [18–22]. Usual physical therapy interventions targeting motor function, particularly balance, gait, and postural control, have demonstrated meaningful clinical benefits in individuals with PD [19,21–23]. In recent years, digital health technologies such as virtual reality (VR) have emerged as promising tools to enhance rehabilitation outcomes [24]. VR-based interventions enable patients to engage in interactive, task-specific exercises within a controlled virtual environment, stimulating both cognitive and motor functions. These systems vary in their level of immersion: non-immersive VR (NIVR) typically involves screen-based interactions with limited sensory feedback; semi-immersive VR (SIVR) offers greater spatial interaction while maintaining awareness of the real world; and immersive VR (IVR) provides a fully immersive experience via head-mounted displays and motion sensors, allowing real-time interaction with a 3D virtual environment [25]. Beyond its technological characteristics, IVR may influence balance control through specific sensorimotor and cognitive mechanisms. Balance impairments in PD are not only related to musculoskeletal deficits but also to altered sensory integration and delayed motor responses to external perturbations. In this context, Campo-Prieto et al. [26] reported that reaction time measured in immersive virtual environments was associated with fall risk in people with PD. These findings suggest that IVR tasks challenging rapid visuomotor responses and sensorimotor integration may target processes that are also fundamental to postural control and fall prevention [26].

Evidence also suggests that VR interventions can improve PD rehabilitation by enhancing balance and gait speed, as well as improving motor function and quality of life [13,27–30].

Despite these promising findings, research specifically evaluating the impact of IVR on static and dynamic balance in PD remains limited [31]. Therefore, the primary aim of this study is to evaluate the effects of an IVR physical therapy program on static and dynamic balance improvement in individuals with PD. Additionally, it will explore its effects on gait speed, functional mobility, lower limb strength, and overall quality of life. Furthermore, potential adverse events associated with IVR use and participants' satisfaction with the intervention will be monitored.

## Materials and methods

### Study design

This protocol has been designed according to the Standard Protocol Items: Recommendations for Interventional Trials (SPIRIT). Two parallel interventions will be compared: an experimental group receiving an IVR physical therapy program to improve static and dynamic balance in different tasks in people with PD, and a control group receiving a usual physical therapy balance exercise program.

Recruitment for this randomized controlled trial began on February 5, 2025. Participant recruitment is ongoing and is expected to be completed at the end of February 2026. Data collection is anticipated to be completed by May 2026, and publication of results is planned by the end of 2026 (Fig 1 and Fig 2). This trial protocol has been reviewed and approved by the Ethics Committee of Aragón (reference number: PI24/501) and prospectively registered at ClinicalTrials.gov (NCT07274514).

Figs 1 and 2 illustrate the design of the clinical trial.

| | Enrollment | | Post-randomization 1-16w | | | Close-out |
|---|---|---|---|---|---|---|
| **TIMEPOINT** | $-t_i$ to 0 | 0 | $1^{th}$ | $4^{th}$ | $16^{th}$ | *After 16 weeks* |
| | | | | | | |
| STUDY DESIGN: | | | | | | |
| Ethic Committee | X | | | | | |
| ENROLLMENT: | | | | | | |
| *Eligibility screen* | | X | | | | |
| *Informed consent* | | X | | | | |
| *Allocation* | | X | | | | |
| INTERVENTION/ COMPARATOR: | | | | | | |
| *Balance physical therapy IVR program* | | | ⟶ | | → | |
| *Usual physical therapy balance exercises program* | | | ⟶ | | → | |
| ASSESSMENTS: | | | | | | |
| *Socio-demographic data* | | | X | | | |
| *Age* | | | X | | | |
| *Sexe* | | | X | | | |
| *Diagnosis date* | | | X | | | |
| *Comorbidities* | | | X | | | |
| *Number of falls* | | | X | X | X | |
| *Balance (Mini-BESTest)* | | | X | X | X | |
| *Functional Mobility (TUG/TUG-DT)* | | | X | X | X | |
| *Gait Speed (10MWT)* | | | X | X | X | |
| *Lower limbs strength (30CST)* | | | X | X | X | |
| *Quality of life (PDQ-39)* | | | X | X | X | |
| *Adverse Symptoms (SSQ)* | | | X | X | | |
| *Satisfaction (Likert scale)* | | | | X | | |
| *Adherence (Attendance record)* | | | | X | | |
| *Statistical Analysis* | | | | | | X |

**Fig 1. SPIRIT schedule enrollment.**

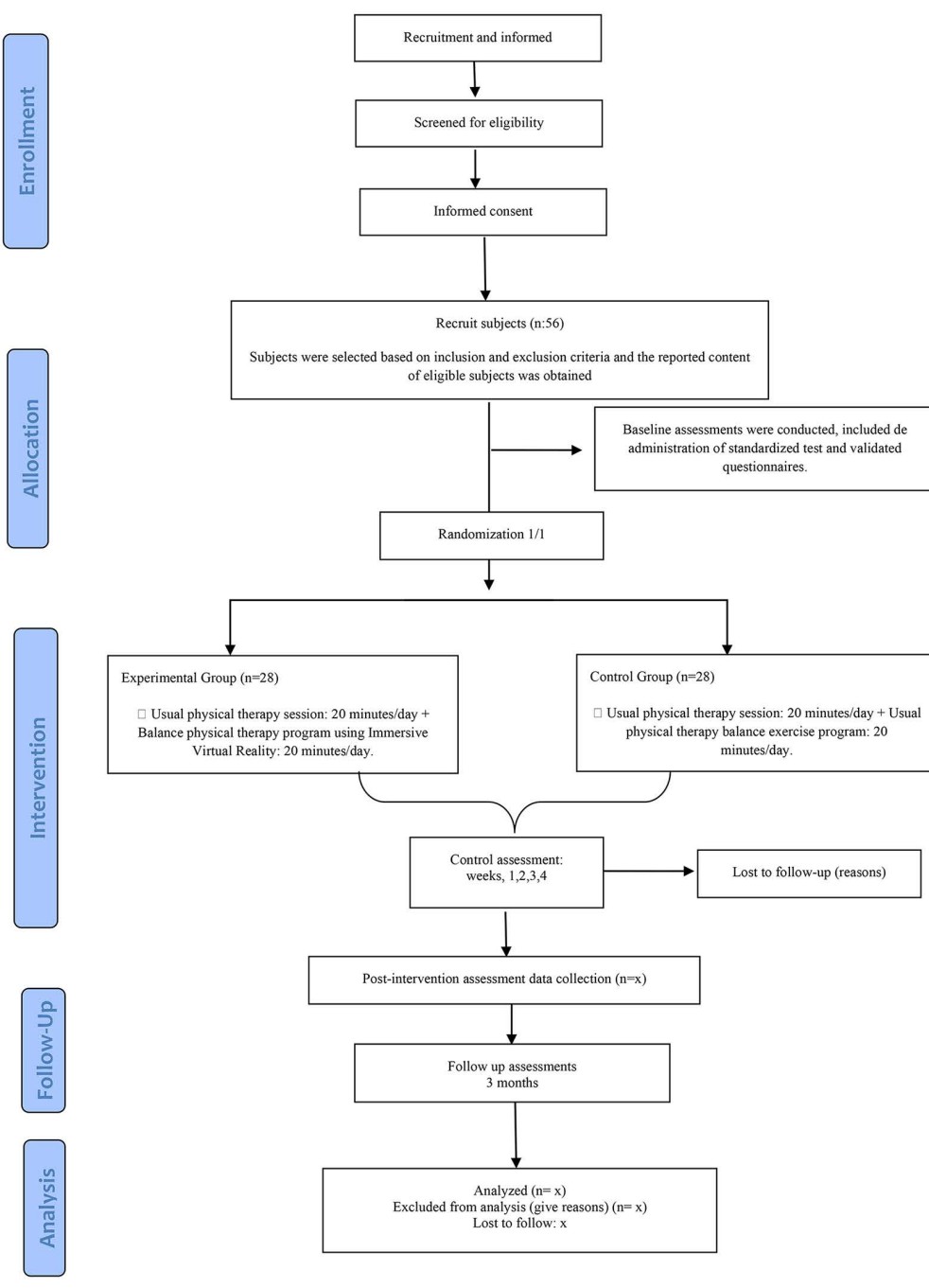

**Fig 2. Flowchart.**

## Setting and population

The study will be conducted with people diagnosed with PD at the Aragon Parkinson's Association in Zaragoza, Spain. Recruitment will be carried out by the physical therapy association team, who will identify eligible candidates during their physical therapy sessions.

## Eligibility criteria

Before inclusion in the study, all potential participants will be fully informed about the study and asked to provide written informed consent. The inclusion criteria will be: 1) Patients diagnosed with PD by a neurologist; 2) Aged 55–90 years and classified as stages 2,5–4 on the modified Hoehn & Yahr scale; 3) Independent in standing and walking, with or without technical aids; 4) A score >24, as evaluated using the validated Spanish version of the Mini-mental State Examination (MMSE-MEC) [32]. The exclusion criteria included will be: 1) Visual or auditory disturbances that make it difficult to perceive the information provided or may cause vertigo or epilepsy; 2) Musculoskeletal disorders or pathological conditions limiting the ability to perform exercises; 3) Psychiatric disorders such as moderate or severe depression, as well as impulse control disorders diagnosed in the last year. The withdrawal criteria will be: 1) Occurrence of severe adverse effects during IVR use, defined as a Simulator Sickness Questionnaire (SSQ) score greater than 20; 2) Voluntary withdrawal by the participant.

## Allocation and blinding

Recruited patients will be randomly assigned to one of the two groups after the baseline evaluations. The sequence will be randomized, with a 1:1 allocation using randomization software (www.randomizer.org) between the experimental and the control group. Each participant will receive an identification code to ensure anonymity. Randomization will be conducted by an independent researcher who is not involved in the treatment or assessment. Outcome assessments will be performed by an evaluator blinded to group assignment, and the research team delivering the intervention will be excluded from all assessment activities to ensure a single-blind design.

## Procedure

Once potential participants are recruited by the physical therapy association team, the research team will provide them with detailed information about the study's objectives. Individuals who potentially meet the inclusion and exclusion criteria will be preselected for participation. Participants who agree to participate will be asked to provide written informed consent. A trained researcher with expertise in neurological assessment will schedule each enrolled participant for the baseline evaluation. All assessments will be conducted by the same evaluator and in the same facility at the Parkinson's Association of Aragón to ensure consistency in testing conditions. Each evaluation session will last approximately 60 minutes.

After the initial assessments, participants will be randomly assigned to either the experimental or control group. Participants will be instructed not to disclose their group assignment to the evaluator during follow-up assessments to minimize potential bias. Intervention in both groups will be administered by 3 physical therapists, each with at least 2 years of experience managing patients with neurological conditions. To ensure consistency and adherence to the study protocol, all therapists will attend a training session focused on standardizing the intervention procedures. Additionally, they will be provided with detailed written guidelines for the intervention. This process will be implemented to ensure consistent patient management across physical therapists and to facilitate accurate documentation of any adverse events that may occur during the intervention.

## Intervention

A total of 8 face-to-face physical therapy sessions will be conducted over a 4-week period, with a frequency of 2 sessions per week and a follow-up 3 months after the intervention. Each session will last 40 minutes, divided into 2 separate 20-minute segments.

The first part of the intervention will be the same for both groups and will consist of a usual physical therapy session based on the Modified Fitness Counts (mFC) exercise program [33], during which participants will perform 15 repetitions of eccentric exercises targeting antigravity muscle groups, including the trunk, knee flexors and extensors, and hip extensors. Exercises will be performed either in a sitting or standing position, depending on the specific muscle group involved. Movements will be executed at a slow contraction velocity to minimize compensatory strategies. The duration of the eccentric phase will be a maximum of 3 seconds and 1 second for the isometric hold [34].

The second part of the treatment protocol will differ according to group allocation. In the experimental group, the second part of the intervention will consist of a personalized static and dynamic balance physical therapy program delivered through IVR using the Oculus Quest 2 VR headset and FisioVR® software (Abaco Digital, Zaragoza, Spain). This program includes a series of 360° spherical images depicting urban environments familiar to the participants, designed to simulate realistic scenarios in which various postural activities are performed. The immersive setting allows participants to move and explore the virtual environment with greater freedom, engaging in both static and dynamic balance mechanisms. Throughout the intervention, the physical therapist will continuously monitor on a computer the exact content the patient is viewing in the VR environment.

At the beginning of each session, participants will be seated comfortably while the VR headset is fitted and adjusted to ensure optimal focus and comfort. A brief familiarization phase will follow, during which participants will identify objects and landmarks within the environment to confirm adequate visual clarity. Throughout this initial phase, subjective feedback regarding visual distortion, discomfort, or disorientation will be collected to monitor adaptation to the immersive environment. This phase will last 3 minutes [35].

The intervention will include 4 distinct virtual scenarios, each designed to challenge different types of balance (static or dynamic) and stability tasks. Depending on task demands, balance can be classified as quiet (in a stable environment), reactive (in response to unexpected perturbations), or proactive (in anticipation of destabilizing forces) [17].

Participants will complete dual tasks, combining balance maintenance with simultaneous motor and/or cognitive challenges (e.g., mental counting, working memory tasks, or internal arithmetic calculations). In each of the 8 sessions, participants will encounter 4 scenarios, each with different exercises. Each exercise will be structured into 4 difficulty levels: Beginner (Level 1), Easy (Level 2), Intermediate (Level 3), and Advanced (Level 4). For each exercise, participants will begin at the Beginner level. Performance will be assessed based on balance. If the exercise is completed with no more than one loss of balance, participants will progress sequentially to the next difficulty level (Easy, Intermediate, and Advanced). This stepwise progression will continue until the highest difficulty level that can be safely performed is reached. In each scenario, participants will perform the exercises at the highest difficulty level achieved according to this criterion. The exercises are shown in Table 1.

Participants in the control group will follow a static and dynamic balance program based on usual physical therapy balance exercises [36]. The specific balance exercise program and its progression will be adapted to each participant's individual condition. As in the experimental group, the usual balance exercises will be structured into 4 progressive levels of difficulty, from Beginner (Level 1) to Advanced (Level 4), with specific exercises adapted to each level. Participants will perform the exercises at the highest possible level of difficulty, which will be assessed by measuring the loss of balance. Participants should not experience more than one loss of balance before progressing to a higher level of difficulty. These exercises are listed in Table 2.

To ensure participant safety and prevent falls, all intervention sessions will be conducted under continuous one-to-one supervision by an experienced physical therapist. The physical therapist will remain nearby at all times, will monitor task execution, and will provide immediate physical assistance in the event of balance loss.

For participants in the experimental group, the physical therapist will administer the SSQ immediately at the end of each session to monitor potential adverse effects associated with IVR exposure, including nausea, oculomotor discomfort, dizziness, and disorientation. SSQ scores and any reported symptoms will be systematically recorded. If relevant symptoms are detected, the session will be paused, and persistent or severe symptoms will result in discontinuation of the intervention and documentation as adverse events. Any adverse events occurring in the control group will be also monitored and reported during the study.

## Outcome measures

**Primary outcome. Balance**. Balance performance will be assessed using the Mini-Balance Evaluation Systems Test (Mini-BESTest). This tool evaluates both static and dynamic balance impairments and identifies the risk of falls. It includes

**Table 1.** Experimental group. Balance physical therapy program using IVR.

| ENVIRONMENTS | | PROGRESSION LEVELS | | | | DUAL TASK |
|---|---|---|---|---|---|---|
| | | BEGINNER | EASY | INTERMEDIATE | ADVANCED | |
| Environment 1. | | Walk around the stage. | Move around the stage by taking a stride. | Move around the stage walking in a semi-tandem gait. | Move around the stage walking in a tandem gait. | Questions will be asked such as:<br>- Look across and tell me what you see.<br>- What shape is the building?<br>- What color is the building?<br>- What book is there on the lectern? |
| Environment 2. | Static balance. | Two-point support, following with eyes and head. | Tandem supports different feet and tracking with eyes and head. | Monopodial support with different feet and tracking with eyes and head. | Monopodial support tracking with eyes and head on unstable surface. | Questions will be asked such as:<br>- Look across and tell me what you see.<br>- What color is the dot?<br>- What color is the wall?<br>- You can point out where the dot is?<br>- Do you hear anything? What? |
| | Dynamic balance. | Walk following the dot, with their normal gait. | Walk following the dot with a striding gait. | Walk following the dot with a semi-tandem gait. | Walk following the dot with a tandem of gait. | |
| Environment 3. | Static balance. | With your feet shoulder-width apart, stand still and try to dodge the balls. | With your feet on the ramp, stand still and try to dodge the balls. | With your feet in tandem, stand still and try to dodge the balls. | With your feet in tandem and on an unstable surface, stand still and try to dodge the balls. | Questions will be asked such as:<br>- What color is the ball?<br>- What kind of ball does it remind you of?<br>- Is that ball used in any sport?<br>- Is there anything else around?<br>- How many balls have come toward you?<br>- How many balls were red?<br>- Do you hear anything? What? |
| | Dynamic balance. | Hit the balls with your hands alternately. 5 times with each hand. | Hit the balls with your feet alternately. 5 times with each foot. | Hit the balls with your hands or feet alternately. 5 times with each of the 4 limbs. | Hit the balls with your hands or feet alternately on an unstable surface. | |
| Environment 4. | | Stay in your position and only dodge people by stepping to the right or left. | Walk, and when you meet someone, avoid them. | Walk with long strides and when you meet someone, avoid them. | Walk in tandem on a line and when you meet someone, avoid them. | Questions will be asked such as:<br>- Does this tunnel seem very dark? Does the light bother you?<br>- Does it change your vision at all?<br>- How many people have you crossed paths with? How many women have you seen? And men?<br>- What color was the first person's shirt? And the last person's pants?<br>- Where are you walking? Describe everything you see.<br>- Describe the girl. Describe the boy. |

14 items grouped into 4 domains: anticipatory postural adjustments, reactive postural control, sensory orientation, and dynamic gait. Each item will be evaluated on a 3-point ordinal scale ranging from 0 to 2, where 0 denotes minimal functional performance and 2 denotes optimal function. The total score can reach a maximum of 28 points, representing the absence of risk of falls [37]. The Mini-BESTest has demonstrated excellent test-retest reliability in individuals with PD [38,39].

**Secondary outcomes. Functional mobility**. Functional mobility will be assessed using the Timed Up and Go test (TUG). From a seated position in a chair with armrests, participants will be instructed to stand up, walk 3 meters, turn around, return to the chair, and sit down [40]. A result of more than 12 seconds will be indicative of a risk of falls [41]. The TUG has demonstrated high interrater reliability in individuals with PD [42].

To assess mobility under cognitive load, the Timed Up and Go Dual Task (TUG-DT) will be used. Participants will follow the same procedure as in the TUG, but while walking, they will simultaneously perform a cognitive task (e.g., counting backward from 70 in steps of 3). A result of more than 14,7 seconds will be indicative of a risk of falls [41]. This test will evaluate functional mobility in dual-task conditions [43].

Table 2. Control group. Usual physical therapy balance exercise program.

| EXERCISE | BALANCE COMPONENT | PROGRESSION PRINCIPLE | DESCRIPTION | PROGRESSION LEVEL | | | |
| --- | --- | --- | --- | --- | --- | --- | --- |
| | | | | BEGINNER | EASY | INTERMEDIATE | ADVANCED |
| Exercise 1. Posture | Static balance | Increase challenge via eyes closed, unstable surface | Place both feet aligned with a separation of 5 cm. Look at a fixed point in front of you. You must maintain this posture without moving for 30 seconds. | Two-foot balance on a stable surface with eyes open. | Two-foot balance on a stable surface with eyes closed | Two-foot balance on an unstable surface with eyes open. | Two-foot balance on unstable surface with eyes closed. |
| Exercise 2. Tandem/ Semi-tandem | Static balance | Increase base of support complexity and sensory challenge | **Semi-tandem:** Place your right heel in front of the sides of your left foot. Hold this position for 30 seconds. Now place your left foot in front and repeat the exercise. Repeat each side as many times as necessary until you change levels. **Tandem**: Place your right heel in front, touching the toes of your left foot. Hold this position for 30 seconds. Now place your left foot in front and repeat the exercise. Repeat each side as many times as necessary until you change levels. | Semi-tandem position with eyes open. | Semi-tandem position with eyes closed. | Tandem position with eyes open. | Tandem position with eyes closed. |
| Exercise 3. Single-leg support | Static balance | Gradual load on single leg + sensory challenge | Place your hands on your hips, then lift your right leg without resting it on the other leg (single-leg support with the non-supporting leg brought forward). Hold this position for 30 seconds. Repeat with each leg as many times as necessary until you change levels. | Single-leg stance with eyes open. | Single-leg stance with eyes closed. | Single-leg stance with eyes open on an unstable surface. | Single-leg stance with eyes closed on an unstable surface. |
| Exercise 4. Inclined ramp | Static balance | Surface instability + eyes open/closed | Climb onto the inclined ramp with your heels touching the floor. Spread your feet shoulder-width apart. Hold this position for 30 seconds. As a precaution, do this near a wall so that the individual can stabilize themselves if necessary. Repeat as many times as necessary until changing levels. | Inclined ramp with eyes open. | Inclined ramp with eyes closed. | Perform the same exercise with your eyes open on an unstable surface. Place foam rubber on the inclined ramp. | Perform the same exercise with your eyes closed on an unstable surface on the inclined ramp. |
| Exercise 5. Compensatory correction step | Dynamic balance | External perturbation, maintain stability | Remain upright, the physical therapist will place their hands against you, one on each shoulder, and you should lean back onto the physical therapist. When the physical therapist removes their hands, you should maintain your balance by taking a step to keep it. Repeat as many times as necessary until you change levels. | Individual with eyes open. | Individual with eyes closed. | Individual on unstable surface with eyes open. | Individual on unstable surface with eyes open. |
| Exercise 6. Semi-tandem and tandem walking | Dynamic balance | Gait complexity + sensory manipulation | Walking in semi-tandem and tandem positions, the physical therapist will indicate whether you should keep your eyes open or closed. Repeat the exercise as many times as necessary until you reach a new level. | Semi-tandem walking with eyes open. | Semi-tandem walking with eyes closed. | Semi-Tandem walking with eyes open. | Tandem walking with eyes closed. |

*(Continued)*

**Table 2.** (Continued)

| EXERCISE | BALANCE COMPO-NENT | PRO-GRESSION PRINCIPLE | DESCRIPTION | PROGRESSION LEVEL | | | |
| --- | --- | --- | --- | --- | --- | --- | --- |
| | | | | BEGINNER | EASY | INTERMEDIATE | ADVANCED |
| Exercise 7. Striding | Dynamic balance | Increase gait complex-ity + base of support | Striding. The physical therapist will indi-cate whether you should keep your eyes open or closed. Repeat the exercise as many times as necessary until you reach a new level. | Striding with eyes open. | Striding with eyes closed. | Striding following a straight line with eyes opened. | Striding following a straight line with eyes closed. |
| Exercise 8. Walking with pelvic–shoulder dissocia-tion | Dynamic balance | Motor-cognitive dual task + gait challenge | Walking with pelvic-shoulder dissocia-tion with two poles (spikes). The physi-cal therapist will give you instructions on whether to do this with your eyes open or closed. Repeat the exercise as many times as necessary until you reach a new level. | Normal walking with two poles that you will move alternately. | Normal walking with two poles that you will move alternately with your eyes closed. | Walk in semi-tandem with two poles, moving them alternately while following a line with your eyes open. | Walk in semi-tandem with two poles, moving them alternately while following a line with your eyes closed. |

**Gait Speed**. Gait speed will be measured using the 10-Meter Walk Test (10MWT). Participants will be instructed to walk 10 meters on a flat surface, performing 3 trials at a comfortable walking speed and 3 at a fast-walking speed. The average speed (in meters/second) for each condition will be calculated. Speeds below 1.1–1.2 m/s would indicate a higher risk of falls [44]. This test has shown excellent test-retest reliability at a comfortable pace in individuals with PD [45].

**Lower limb strength**. Lower limb strength will be evaluated using the 30-Second Sit-to-Stand Test (30CST), which is part of the Senior Fitness Test (SFT) [46]. Participants will be seated on a chair with their back straight, feet flat on the floor, and arms crossed over their chest. They will be asked to complete as many full sit-to-stand movements as possible within 30 seconds [47]. The 30CST has demonstrated excellent test-retest reliability in community-dwelling older adults [48].

**Quality of life**. Health-related quality of life will be measured using the Parkinson's Disease Questionnaire-39 (PDQ-39). This self-administered instrument consists of 39 items distributed across 8 dimensions and will assess quality of life during the previous month in individuals with PD. Each item is scored on a 5-point ordinal scale ranging from 0 (never) to 4 (always). The scores for each dimension are averaged and transformed to a scale of 0–100, where 0 represents the best possible quality of life and 100 the worst. Lower values will indicate a better perceived quality of life, and higher val-ues will reflect a greater impact of the disease [49]. The PDQ-39 has demonstrated high test-retest reliability [50].

**Adverse symptoms**. Adverse symptoms related to IVR exposure will be monitored using the Simulator Sickness Ques-tionnaire (SSQ). This questionnaire includes 16 items across 3 domains: oculomotor symptoms, nausea, and disorienta-tion. Symptoms will be scored for intensity and interpreted as follows: < 5 = insignificant, 5–10 = minimal, 10–15 = significant, 15–20 = concerning, and > 20 = severe [51]. This questionnaire will also serve as a criterion for discontinuing participation in the IVR physical therapy program. Participants who score 20 will be withdrawn from the study, as this reflects the presence of severe symptoms identified by the scale. The SSQ has demonstrated good internal consistency [52].

**Satisfaction**. Participant satisfaction with the intervention will be assessed using an *ad hoc* one-question questionnaire developed for this clinical trial. Responses will be recorded on a 5-point Likert scale, where 1 will indicate "very dissatis-fied" and 5 will indicate "very satisfied."

**Adherence**. A participant's adherence will be evaluated based on attendance records during the intervention phase. Individuals were categorized as showing *high adherence* when they attended no fewer than 80% of the scheduled ses-sions and completed all prescribed exercises or more. In contrast, *non-adherence* was defined as attending fewer than 20% of the total sessions planned [53].

## Sample size calculation

Sample size estimation was performed using G*Power software (version 3.1.9.7), taking the primary outcome variable, balance, assessed by the Mini-BESTest scale as the reference. Although the hypothesis anticipates superior outcomes for the experimental IVR intervention compared to the control condition, a two-tailed alternative hypothesis was adopted to maintain a conservative statistical approach. The calculation was based on a repeated measures design involving two groups and three time points, but specifically on the expected difference between groups at the post-intervention time point assuming a moderate effect size (d = 0.71), a significance level of 0.05, and a statistical power of 80%. Although the study includes follow-up measurements, this approach provides a conservative estimate expected to offer adequate power for comparisons at follow-up. Because the sample size calculation relies on the expected between-group difference at the post-intervention time point, assumptions about the correlation between repeated measurements will not be necessary. Under these conditions, the required sample size was determined to be 50 participants, with 25 allocated to each group. To accommodate an estimated 10% attrition rate, the final target sample was increased to 56 participants (28 per group).

## Data management

A comprehensive database will be created to monitor each participant's progress and record all assessment data. All collected information will be securely stored on password-protected computers accessible only to the researchers directly involved in the study. An independent investigator will oversee the progress and safety of data collection. Data analysis will be conducted once recruitment and data collection have been completed.

## Statistical analysis

Statistical analysis will be performed with Statistical Package for the Social Sciences version 29.0 (SPSS Inc, Chicago, IL). The significance level will be 0.05 for all statistical analyses. Descriptive statistics will summarize participants' baseline characteristics and study outcomes. Categorical variables will be presented as absolute and relative frequencies (n, %), whereas continuous variables will be expressed as mean ± standard deviation or, when appropriate, as median and interquartile range, together with 95% confidence intervals. The Shapiro–Wilk test will assess the normality of continuous variables. To ensure comparability between groups at baseline, appropriate parametric or nonparametric tests will be applied depending on data distribution: t-tests or Mann–Whitney U tests for continuous data, and Chi-square tests for categorical data.

To examine the effects of the intervention across time, linear mixed-effects models will be employed to analyze group-by-time interactions across the three measurement points. Significant main or interaction effects will be further explored through Bonferroni-adjusted post hoc comparisons. In cases where data deviate from normality, within-group changes will be analyzed using the Friedman test, followed by Wilcoxon signed-rank tests for pairwise comparisons. Bonferroni correction will be applied to control for multiple testing and Type I error.

Between-group differences at each time point will be examined using independent-samples t-tests (accompanied by Levene's test for homogeneity of variances) for normally distributed data, or Mann–Whitney U tests for non-parametric data. Categorical outcomes will be compared using Chi-square or Fisher's exact test, as appropriate.

All analyses will follow the intention-to-treat (ITT) principle, including every randomized participant in the group to which they were initially assigned, regardless of adherence, or protocol deviations. Missing data will be managed through suitable statistical procedures, such as last observation carried forward (LOCF) or multiple imputation, depending on the type and extent of missingness.

Effect sizes will be computed to determine the magnitude of observed differences, using Cohen's d for parametric tests and effect size r for non-parametric analyses, both within and between groups for the primary outcomes. Effect sizes will

be reported with 95% confidence intervals, and interpretation will consider both statistical significance and clinical relevance, using established Minimal Clinically Important Differences (MCIDs) for all outcomes. Insignificant, small, medium, and large differences will be reflected in effect sizes of < 0.2, 0.2–0.5, 0.5–0.8, and > 0.8, respectively [54].

**Ethical aspects and dissemination**

The study will be conducted in accordance with the ethical principles outlined in the Declaration of Helsinki. The protocol has been approved by the Aragón Ethics Committee (reference number: PI24/501); current version dated 11,25, 2024.

Access to the complete dataset will be restricted to the researcher responsible for the statistical analysis. To maintain participant confidentiality, all data will be coded and anonymized, ensuring that no identifying information can be linked to individual participants.

Upon study completion and analysis finalization, the open-access repository Zenodo will house the entirety of the raw quantitative data, codebooks, and statistical analysis scripts. The findings of this study will be disseminated through several channels to maximize their impact. Primary dissemination will occur through publication in peer-reviewed scientific journals, with PLOS ONE being the targeted journal. Presentations at national and international conferences related to physical therapy and health care will also be pursued. Furthermore, the results will be shared with participating in healthcare institutions, associations of patients with neurological disorders and, if applicable, with the study participants.

After the publication of the findings, the study's data will be made available upon request, with full respect for the confidentiality of all participants.

## Discussion

This randomized controlled trial protocol has been designed to analyze the effects of an IVR physical therapy program on balance improvement in individuals with PD. The primary objective of this study will determine whether the IVR intervention produces improvements in both static and dynamic balance in people with PD. In addition, this protocol includes several secondary variables that may indicate broader functional improvements, such as gait speed, functional mobility, overall lower limb strength, functional mobility, and quality of life. Adverse effects related to IVR exposure, participant satisfaction, and adherence will also be essential for evaluating the acceptability of the intervention.

Regarding balance, growing evidence on the effects of VR suggests that IVR interventions constitute a promising therapeutic strategy for improving both balance and gait performance in individuals with PD. Notably, Goffredo et al. reported clinically meaningful improvements in dynamic balance, as shown by Mini-BESTest scores following NIVR training [55]. Consistent findings have been reported by Kashif et al. and Gulcan et al., who observed significant improvements in balance measured with the Berg Balance Scale (BBS) [20,56]. Likewise, Esculier et al. reported improvements in balance using the Tinetti scale after NIVR environments, further supporting the potential role of VR as a tool for balance physical therapy rehabilitation for this population [27]. Regarding SIVR approaches, Tunur et al., demonstrated in their pilot study that a 3-week intervention using Moving Through Glass (MTG) improved balance, as measured by the Mini-BESTest [57]. Pullia et al. also identified significant post-intervention improvements in BBS scores [58]. In relation to IVR, Cano Porras et al. reported a clinically meaningful improvement in balance among patients with PD following 12 sessions of IVR, resulting in an increase of 2.47 points on the Mini-BESTest [59]. Honzíková et al. demonstrated that 8 sessions of IVR over a period of 1 month, resulted in improved balance, reflected by a 2-point increase on the BBS [60]. Moreover, Brandín-de la Cruz et al. demonstrated that combining IVR with antigravity treadmill training led to substantial improvements in Tinetti scale scores [61], suggesting that multimodal IVR-based protocols may enhance therapeutic benefits. Similarly, Yun et al. reported improved balance performance measured by the BBS after an IVR intervention integrated with physical and cognitive training tasks [62]. These findings collectively highlight the potential of VR programs for addressing postural instability in PD.

On the other hand, the evidence regarding the effects of VR on functional mobility in PD shows mixed results. Some studies have reported significant improvements in TUG performance after SIVR-based interventions, suggesting benefits for functional mobility [13,63]. However, findings for TUG-DT are more inconsistent, with some studies showing no

significant changes in participants who received IVR sessions [60,62]. Such variability underscores the need for further research into the specific features of IVR that may enhance mobility, particularly under dual-task conditions.

Regarding gait speed, Honziková et al. demonstrated that IVR training significantly reduced the time required to complete the 10MWT [60]. For SIVR programs, Gandolfi et al. used a program consisting of 21 sessions over 7 weeks [64], and Domínguez et al. implemented a program over 8 weeks, with 3 sessions per week [65]; both studies showed meaningful improvements in gait performance as measured by the 10MWT.

Considering lower limb strength, Lee et al. [66] reported significant improvements in the Sit-to-Stand test in the experimental group that underwent the NIVR training program compared to the group that received usual physical therapy.

In relation to quality of life, an article reports that people with PD who participated in NIVR interventions experienced greater improvements in quality of life than those who received usual physical therapy care, with a 10% increase in PDQ-39 scores [67]. Triegaardt et al., [68] in their systematic review and meta-analysis, supported these findings by showing that VR interventions led to significant clinical improvements in the quality of life of patients with PD, as assessed by the PDQ-39 questionnaire.

Adverse effects potentially associated with the use of IVR, such as nausea, dizziness, blurred vision, and spatial disorientation, collectively referred to as cybersickness [35,69] will be assessed using the SSQ scale. In individuals with PD, these symptoms may be exacerbated due to impairments in the integration of visual and somatosensory information [35]. However, the extent to which these integration deficits affect susceptibility to cybersickness in PD remains underexplored. Pimenta Silva et al. [70] reported that at least one adverse symptom occurred in 8.5% of the 1012 sessions conducted among participants who underwent VR. However, they noted that only 20% of these symptoms were directly related to VR.

The therapeutic potential of IVR may be partly attributed to the multimodal sensory input it provides, which could facilitate motor learning by engaging alternative neural pathways. This sensory enrichment may be especially relevant for individuals with PD, where sensorimotor integration is often disrupted. Evidence from neuroimaging studies indicates that IVR-based tasks can activate cortical areas involved in action observation and execution, particularly within frontal and parietal regions associated with mirror neuron networks, potentially supporting both cognitive and motor aspects of rehabilitation [25,71].

In addition, IVR has been associated with greater patient engagement compared to usual physical therapy, helping to sustain motivation throughout the intervention. Some studies also suggest that it may contribute to cognitive improvement [72], which in turn could support long-term adherence to treatment [18,31].

## Strengths and limitations

This study offers strengths, including a novel intervention using an IVR physical therapy program based on levels and dual tasks, which has not yet been implemented in this population. However, the study has some limitations. One limitation is its single-blind design, as participants will be aware of their group allocation, potentially introducing performance bias. To mitigate this, all participants will receive equal attention and session time, and outcome assessors will remain blinded to group assignments to reduce detection bias. Another limitation is the inability to isolate the specific effects of the IVR intervention, since, for ethical reasons, it must be delivered alongside any ongoing individual or group therapies participants are receiving. This will be managed by carefully documenting all concurrent treatments to allow for appropriate adjustments during data analysis. Additionally, there is a potential risk of adverse effects associated with IVR use (such as dizziness or visual discomfort), which could lead to participant withdrawal. To minimize this risk, participants will be closely monitored during sessions, and protocols will be in place for early detection and management of side effects, including gradual adaptation periods and optional breaks during VR exposure.

## Author contributions

**Conceptualization:** Lorena Morcillo-Martínez, Sandra Calvo, Natalia Brandín-de la Cruz.

**Data curation:** Lorena Morcillo-Martínez.

**Formal analysis:** Carolina Jiménez-Sánchez.

**Investigation:** Lorena Morcillo-Martínez, María Sanz-Mármol, Adriana Isabel Romanos-Navarrete, Sandra Calvo, Lilian Le Roux-Ethève, Laura Esteban-Repiso, Paula Córdova-Alegre, Carolina Jiménez-Sánchez, Aitor Garay-Sánchez, Natalia Brandín-de la Cruz.

**Methodology:** Lorena Morcillo-Martínez, Sandra Calvo, Natalia Brandín-de la Cruz.

**Supervision:** Sandra Calvo, Natalia Brandín-de la Cruz.

**Validation:** Sandra Calvo, Natalia Brandín-de la Cruz.

**Writing – original draft:** Lorena Morcillo-Martínez.

**Writing – review & editing:** María Sanz-Mármol, Adriana Isabel Romanos-Navarrete, Sandra Calvo, Lilian Le Roux-Ethève, Laura Esteban-Repiso, Paula Córdova-Alegre, Carolina Jiménez-Sánchez, Aitor Garay-Sánchez, Natalia Brandín-de la Cruz.

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
