## [Decision Letter · Decision Letter 0]

8 Dec 2025

Dear Dr. Calvo,

Thank you for submitting your manuscript to PLOS ONE. After careful consideration, we feel that it has merit but does not fully meet PLOS ONE’s publication criteria as it currently stands. Therefore, we invite you to submit a revised version of the manuscript that addresses the points raised during the review process.

The study protocol, while timely and clinically relevant, requires substantial revisions to address several conceptual and methodological issues. Key concerns include the lack of explicit justification for focusing solely on balance and failing to distinguish between static and dynamic balance. The rationale must be strengthened by integrating recent evidence to clarify the added value of IVR. For reproducibility, the intervention description must strictly apply the TIDieR Checklist, detailing exercise progression criteria, the underlying theoretical model, and specific parameters. Critically, the protocol must specify robust safety monitoring procedures for adverse effects like cybersickness, clear discontinuation criteria, and fall procedures. Methodologically, the sample size's effect size must be justified, adherence rules must be explicitly stated, and detail feasibility goals must be included among others. Kindly refer to the reviewer's comments below for further details. The major issue that the are missing details that make it difficult  critically evaluate the intervention at this stage.

We look forward to receiving your revised manuscript.

Kind regards,

Domna Banakou

Academic Editor

PLOS One

2. Please update the manuscript to provide details of the Clinical Trial registration. If the government shutdown is still ongoing, please highlight this in your Response to Reviewers.

3. Please include your tables as part of your main manuscript and remove the individual files. Please note that supplementary tables (should remain/ be uploaded) as separate "supporting information" files.

4. In the online submission form, you indicated that [No datasets have been generated or analyzed yet, as this is a study protocol. Upon completion of the trial, the data will be available from the corresponding author on reasonable request, in accordance with privacy and data protection regulations.

Upon study completion and analysis finalization, the open-access repository Zenodo will house the entirety of the raw quantitative data, codebooks, and statistical analysis scripts. The findings of this study will be disseminated through several channels to maximize their impact. Primary dissemination will occur through publication in peer-reviewed scientific journals, with PLOS ONE being the targeted journal. Presentations at national and international conferences related to physical therapy and health care will also be pursued. Furthermore, the results will be shared with participating in health-care institutions, associations of patients with neurological disorders and, if applicable, with the study participants.

After the publication of the findings, the study’s data will be made available upon request, with full respect for the confidentiality of all participants.].

5. Please include a copy of Table 1-2 which you refer to in your text on page 10.

Additional Editor Comments:

The protocol is timely and clinically relevant, but several revisions are needed before re-submission. The literature review motivates VR for PD balance yet should more explicitly justify why immersive VR yields incremental benefits and acknowledge cybersickness risks up front. Design is sensible, but feasibility needs concrete recruitment targets and task thresholds. Methods require cleanup as there are eligibility mismatches (2.5–4 vs 1–4), the authors should harmonize adherence/withdrawal rules, detail safety monitoring (SSQ timing per session, AE/SAE definitions, fall procedures), and ensure intervention reproducibility via fully specified exercise tables, therapist training, and fidelity checks. The analysis plan should pre-specify linear mixed-effects models for time x group, control multiplicity (e.g., BH-FDR for secondaries), report effect sizes with CIs, and interpret against MCIDs.

Reviewers' comments:

Reviewer's Responses to Questions

**Comments to the Author**

1. Does the manuscript provide a valid rationale for the proposed study, with clearly identified and justified research questions?

Reviewer #1: Yes

2. Is the protocol technically sound and planned in a manner that will lead to a meaningful outcome and allow testing the stated hypotheses?

Reviewer #1: Partly

3. Is the methodology feasible and described in sufficient detail to allow the work to be replicable?

Reviewer #1: No

4. Have the authors described where all data underlying the findings will be made available when the study is complete?

Reviewer #1: No

5. Is the manuscript presented in an intelligible fashion and written in standard English?

Reviewer #1: Yes

You may also provide optional suggestions and comments to authors that they might find helpful in planning their study.

Reviewer #1: General Comments

The aim of the study is to evaluate the effects of an immersive virtual reality (IVR) program with balance exercises compared to conventional balance exercises without IVR, with balance as the primary outcome. However, some conceptual and methodological issues need clarification and improvement to strengthen the protocol.

1. Title and Conceptual Focus

The title states that the trial focuses on balance, but the authors do not justify in the abstract or introduction why balance is the central and only outcome highlighted.

2. Missing Rationale and Recent Evidence

The introduction lacks a critical justification of the importance of balance training in Parkinson’s disease and how IVR may modulate specific mechanisms related to balance.

Recent studies should be incorporated to contextualize the protocol within current evidence. In particular:

De Natale et al. (2025), which provides updated insights into the effects of immersive VR in PD and helps identify remaining gaps in the literature.

Campo-Prieto et al. (2023), which explored reaction time in IVR as a predictor of falls and adds conceptual depth to the potential mechanisms by which IVR may influence balance and fall risk.

Including these papers will help the authors clarify what remains uncertain in the literature and justify the added value of publishing this protocol.

3. Intervention Description (TIDieR Checklist Needed)

The rationale for publishing a protocol is to deliver a transparent and reproducible description of the intervention. However, essential aspects are missing.

Authors should apply the TIDieR Checklist, as recommended for clinical trial protocols, to detail:

Whether the protocol targets static balance, dynamic balance, or both. It is unclear whether the authors are referring to static balance, dynamic balance, or both. This distinction should be explicit in the title, abstract, and introduction, given that Parkinson’s disease affects multiple dimensions of balance.

The specific exercises must include: progression criteria and the theoretical model behind progression (e.g., motor learning principles) and exercise parameters according to the FITT-VP model.

Without these details, it is difficult for clinicians and researchers to reproduce or critically evaluate the intervention.

4. Safety Considerations

VR interventions frequently raise concerns regarding dizziness, cybersickness, imbalance, and discomfort. The protocol does not explain:

How the research team will monitor adverse effects?

Criteria for discontinuation or adaptation?

Whether patients will perform tasks while standing, and how safety will be ensured?

These are critical elements for feasibility and ethical adequacy.

5. Sample Size Justification

The procedure used to calculate sample size is unclear.

The authors should justify:

Why a specific effect size was chosen?

Whether it was based on previous VR studies in PD?

A transparent sample size description is essential for protocol validity.

**Do you want your identity to be public for this peer review?** For information about this choice, including consent withdrawal, please see our Privacy Policy

Reviewer #1: No

---

## [Author Response · Author response to Decision Letter 1]

2 Jan 2026

We appreciate your review of our manuscript. We also welcome your comments to improve the quality of the manuscript. We have made the requested changes and revised the manuscript accordingly. Attached is a point-by-point response. We hope you find our responses satisfactory.

Editor comments:

The study protocol, while timely and clinically relevant, requires substantial revisions to address several conceptual and methodological issues. Key concerns include the lack of explicit justification for focusing solely on balance and failing to distinguish between static and dynamic balance. The rationale must be strengthened by integrating recent evidence to clarify the added value of IVR. For reproducibility, the intervention description must strictly apply the TIDieR Checklist, detailing exercise progression criteria, the underlying theoretical model, and specific parameters. Critically, the protocol must specify robust safety monitoring procedures for adverse effects like cybersickness, clear discontinuation criteria, and fall procedures. Methodologically, the sample size's effect size must be justified, adherence rules must be explicitly stated, and detail feasibility goals must be included among others. Kindly refer to the reviewer's comments below for further details. The major issue that the are missing details that make it difficult critically evaluate the intervention at this stage.

Response:

Thank you for your comments. A detailed, point-by-point response to all comments from the Editor and Reviewer is provided below, indicating the exact changes made and their location in the revised manuscript.

The protocol is timely and clinically relevant, but several revisions are needed before re-submission. The literature review motivates VR for PD balance yet should more explicitly justify why immersive VR yields incremental benefits and acknowledge cybersickness risks up front.

Design is sensible, but feasibility needs concrete recruitment targets and task thresholds. Methods require cleanup as there are eligibility mismatches (2.5–4 vs 1–4), the authors should harmonize adherence/withdrawal rules, detail safety monitoring (SSQ timing per session, AE/SAE definitions, fall procedures), and ensure intervention reproducibility via fully specified exercise tables, therapist training, and fidelity checks.

The analysis plan should pre-specify linear mixed-effects models for time x group, control multiplicity (e.g., BH-FDR for secondaries), report effect sizes with CIs, and interpret against MCIDs.

Response:

In relation to the update of the clinical trials registry, we would like to inform the editor and reviewers that the authors have now obtained the registration code (NCT07274514), following the changes that occurred due to the government shutdown. This registration code has already been included in the manuscript.

In relation to the rationale for IVR and its associated cybersickness risks, we have revised the Introduction and Methods sections to provide more detail on the mechanisms and benefits of IVR for balance rehabilitation in PD, as well as the associated risk of cybersickness. The Introduction now emphasizes that, beyond its technological characteristics, IVR may improve balance by engaging specific sensorimotor and cognitive processes, including rapid visuomotor responses and sensory integration, which are critical for postural control and fall prevention. Evidence from Campo-Prieto et al. (2023) is cited to support this rationale. The Methods section specifies how adverse effects are monitored using the Simulator Sickness Questionnaire (SSQ) after each session, with clear criteria for pausing or discontinuing participation to ensure participant safety.

Concerning the previously identified inconsistencies in eligibility criteria (Hoehn and Yahr stages 2.5–4 vs. 1–4) have been corrected and harmonized throughout the manuscript.

In relation to safety monitoring and falls prevention the Methods section has been expanded to provide a detailed description of safety monitoring procedures. Specifically, we now clarify the timing of the SSQ, which is administered at the end of each intervention session, define how adverse events are monitored and recorded, and describe fall prevention procedures, including continuous one-to-one supervision by an experienced physical therapist and immediate assistance in case of balance loss.

Regarding Intervention reproducibility, to improve reproducibility, the information in the exercises tables has been expanded with specifications, detailing the type of balance as well as the progression criteria and the session structure.

Regarding statistics analysis, thank you for your valuable feedback. We have revised the analysis plan to incorporate your suggestions regarding linear mixed-effects models, reporting effect sizes with confidence intervals, and interpreting results against MCIDs. Specifically, we will use linear mixed-effects models for time × group interactions, report effect sizes with 95% confidence intervals, and interpret findings considering MCIDs for all outcomes. Regarding multiplicity control, we considered BH-FDR; however, given that our secondary outcomes are confirmatory, limited in number, and all comparisons are pre-specified, we will apply Bonferroni correction for post hoc tests. This approach aligns with CONSORT recommendations for trials with a small set of secondary endpoints. We believe these changes strengthen the rigor and transparency of our protocol.

Reviewers' comments:

Reviewer #1: General Comments

The aim of the study is to evaluate the effects of an immersive virtual reality (IVR) program with balance exercises compared to conventional balance exercises without IVR, with balance as the primary outcome. However, some conceptual and methodological issues need clarification and improvement to strengthen the protocol.

1. Title and Conceptual Focus

The title states that the trial focuses on balance, but the authors do not justify in the abstract or introduction why balance is the central and only outcome highlighted.

Response:

Thank you for your comment. We agree that the rationale for emphasizing balance as the primary outcome can be made more explicit. Balance was selected as the central outcome because postural instability is a hallmark feature of Parkinson’s disease (PD) and a major contributor to falls, functional dependence, social isolation, and reduced quality of life. For this reason, balance represents a clinically relevant target for intervention.

While the trial also evaluates gait speed, functional mobility, lower limb strength, and quality of life as secondary outcomes, the title highlights balance to reflect the primary focus of the intervention and its main clinical relevance. To address the reviewer’s concern, we have revised the abstract and introduction to more clearly justify balance as the primary outcome of the study.

2. Missing Rationale and Recent Evidence

The introduction lacks a critical justification of the importance of balance training in Parkinson’s disease and how IVR may modulate specific mechanisms related to balance.

Recent studies should be incorporated to contextualize the protocol within current evidence. In particular:

De Natale et al. (2025), which provides updated insights into the effects of immersive VR in PD and helps identify remaining gaps in the literature. Campo-Prieto et al. (2023), which explored reaction time in IVR as a predictor of falls and adds conceptual depth to the potential mechanisms by which IVR may influence balance and fall risk. Including these papers will help the authors clarify what remains uncertain in the literature and justify the added value of publishing this protocol.

Response:

Thank you for your suggestion to incorporate recent literature to strengthen the conceptual framework of the study. Following this recommendation, we have added the study by Campo-Prieto et al. (2023) to the Introduction to provide additional mechanistic context. Although balance was not the primary outcome in that study, its findings linking reaction time in immersive virtual environments with fall risk support the relevance of sensorimotor integration and rapid motor responses, processes that are also fundamental to postural control in PD. With regard to De Natale et al. (2025), while we acknowledge its value as a recent overview of VR-based interventions in PD, this review predominantly focuses on NIVR, and the IVR studies it discusses are already included in our reference list. Therefore, we considered that its inclusion would not substantially add new evidence or conceptual insight specific to IVR-based balance training beyond what is already covered in the current manuscript.

In addition, we have strengthened the Introduction to explicitly highlight the clinical relevance of postural instability in PD, noting that 45–68% of patients experience falls annually. This high incidence of falls, along with the associated loss of functional independence and social isolation, underscores the importance of targeting balance as the primary outcome. IVR may improve balance by enhancing sensorimotor integration, promoting rapid motor responses, and providing real-time multisensory feedback, all of which are fundamental for postural control in PD. Despite existing studies, there remains a gap in the literature regarding protocols that specifically evaluate the effects of IVR on balance as a primary outcome, which this trial aims to address.

3. Intervention Description (TIDieR Checklist Needed)

The rationale for publishing a protocol is to deliver a transparent and reproducible description of the intervention. However, essential aspects are missing.

Authors should apply the TIDieR Checklist, as recommended for clinical trial protocols, to detail:

Response:

Thank you for suggesting the use of the TIDieR Checklist. We would like to clarify that the manuscript has been prepared in accordance with the CONSORT checklist, which is the most widely used reporting guideline for randomized controlled trials and is explicitly required by PLOS ONE. The use of CONSORT ensures transparent, complete, and standardized reporting of the trial design, conduct, and safety procedures.

In addition, we agree that the TIDieR Checklist is a valuable framework for reporting clinical trial interventions, and its elements are already addressed in the manuscript as follows:

Brief name: Title and Abstract (page 1).

Why: Introduction (paragraphs 2–3).

What (materials): Methods, section “Intervention” (page 9-10).

What (procedures): Methods, section “Intervention” (page 9-10).

Who: Section “procedure” (page 8).

How (mode of delivery): Methods, section “Intervention delivery” (page 8-10).

Where: Added section “Intervention” (page 8).

When and how much: Section “Intervention” (page 8-9).

Tailoring: Section Intervention (Table 1- Table 2) “Progression and adaptation” (page 10-11).

Modifications: Methods (page 10-11).

How well (planned fidelity): Methods (page 13).

How well (actual fidelity): Methods (page 13).

Whether the protocol targets static balance, dynamic balance, or both. It is unclear whether the authors are referring to static balance, dynamic balance, or both. This distinction should be explicit in the title, abstract, and introduction, given that Parkinson’s disease affects multiple dimensions of balance.

Response:

Thank you for your comment. We have revised the title, abstract, and introduction to explicitly state that the protocol targets both static and dynamic balance in people with PD, thereby clarifying the scope of the balance outcomes addressed.

The specific exercises must include: progression criteria and the theoretical model behind progression (e.g., motor learning principles) and exercise parameters according to the FITT-VP model.

Without these details, it is difficult for clinicians and researchers to reproduce or critically evaluate the intervention.

Response:

Thank you for your comment. Progression across difficulty levels is grounded in established motor learning principles and is designed to optimize motor adaptation and balance control while minimizing excessive loss of stability. Exercises are task-specific and progressively increase in complexity through reductions in base of support and augmentation of sensory and cognitive demands, consistent with the challenge point framework. The protocol also incorporates variability of practice, augmented feedback provided by the immersive VR environment, and a distributed practice structure. Progression is structured to promote error-based learning, retention, and transfer of balance skills, while the immersive and interactive nature of the intervention enhances attentional focus, motivation, and engagement.

In relation to the Exercise parameters for the experimental group (FITT-VP):

•Frequency: 2 sessions per week.

•Intensity: Highest level achieved without more than one loss of balance in each exercise.

•Time: 2–3 minutes per exercise.

•Type: Static and dynamic balance exercises, with dual-task demands (cognitive and quiet, proactive and reactive task).

•Volume: 6 exercises per session with four difficulty levels.

•Progression: Increased task complexity, reduced base of support, increased sensory and cognitive load.

Regarding Exercises parameters for the control group (FITT_VP):

•Frequency: 2 sessions per week.

•Intensity: Highest level achieved without more than one loss of balance in each exercise.

•Time: 2–3 minutes per exercise.

•Type: Static and dynamic balance exercises, with dual-task demands (cognitive and quiet, proactive and reactive task).

•Volume: 8 exercises per session with four difficulty levels.

•Progression: Increased task complexity, reduced base of support, increased sensory and cognitive load.

4. Safety Considerations

VR interventions frequently raise concerns regarding dizziness, cybersickness, imbalance, and discomfort. The protocol does not explain:

How the research team will monitor adverse effects?

Response:

Thank you for your comment. Adverse effects related to IVR exposure will be systematically monitored at the end of each intervention session using the SSQ, administered by an experienced physical therapist. The SSQ will be used to assess symptoms such as nausea, dizziness, oculomotor discomfort, and disorientation.

SSQ scores and any reported adverse symptoms will be recorded after each session as part of the safety monitoring procedures. This questionnaire will also serve as a criterion for discontinuing participation in the VR physiotherapy program. Participants obtaining a score of 20 will be withdrawn from the study, as this threshold reflects the presence of severe symptoms according to the scale. In addition, if clinically relevant symptoms are identified during a session, the intervention may be paused to ensure participant’s safety. These clarifications have been incorporated into the Methods section.

Criteria for discontinuation or adaptation?

Response:

Thank you for your comment regarding criteria for discontinuation or adaptation. At the beginning of each session, participants will undergo a brief 3-minute familiarization phase while seated comfortably, during which the VR headset is adjusted to ensure optimal focus and comfort. During this phase, subjective feedback regarding visual distortion, discomfort, or disorientation will be collected to assess initial adaptation to the immersive environment.

If mild or transient symptoms are reported, the intervention may be adapted by allowing additional rest time, readjusting the VR equipment, or reducing exposure as needed. In addition, the SSQ will be administered at the end of each session. Participants who score 20 on the SSQ will be withdrawn from the study, as this threshold reflects the presence of severe symptoms according to the scale. In addition, the physical therapist will always be attentive to the patient, inviting them to take breaks if necessary.

Together, these procedures constitute predefined and objective criteria for adapting or discontinuing the intervention to ensure participant’s safety.

Whether patients will perform tasks while standing, and how safety will be ensured?

Response:

We would like

---

## [Decision Letter · Decision Letter 1]

21 Jan 2026

Dear Dr. Calvo,

Thank you for submitting your manuscript to PLOS ONE. After careful consideration, we feel that it has merit but does not fully meet PLOS ONE’s publication criteria as it currently stands. Therefore, we invite you to submit a revised version of the manuscript that addresses the points raised during the review process.

We look forward to receiving your revised manuscript.

Kind regards,

Domna Banakou

Academic Editor

PLOS One

Journal Requirements:

Additional Editor Comments:

As part of the PLOS ONE protocol review process, the paper was reviewed by a statistical reviewer. Please refer carefully to the reviewer’s comments and address the minor concerns raised.

Reviewers' comments:

Reviewer's Responses to Questions

**Comments to the Author**

1. Does the manuscript provide a valid rationale for the proposed study, with clearly identified and justified research questions?

Reviewer #2: Yes

2. Is the protocol technically sound and planned in a manner that will lead to a meaningful outcome and allow testing the stated hypotheses?

Reviewer #2: Yes

3. Is the methodology feasible and described in sufficient detail to allow the work to be replicable?

Reviewer #2: Yes

4. Have the authors described where all data underlying the findings will be made available when the study is complete?

Reviewer #2: Yes

5. Is the manuscript presented in an intelligible fashion and written in standard English?

Reviewer #2: Yes

You may also provide optional suggestions and comments to authors that they might find helpful in planning their study.

Reviewer #2: The authors present the protocol for a single-blind two arm randomized controlled trial in individuals with Parkinson's disease, evaluating immersive virtual reality (IVR) based balance training versus standard balance training. The manuscript will be strengthened if the authors consider the following points.

1. in the Eligibility criteria section, authors mention that adherence to the intervention protocol below 90% is a reason for withdrawal. They later say that analyses will be based on the intent-to-treat principle, which includes all individuals who have been randomized (so adherence is not a factor). These two statements are inconsistent and should be clarified.

2. Some clarification is still warranted for determining the difficulty level of the exercises, particularly for those in the IVR group. Authors state that individuals will perform all exercises at the highest possible difficulty level (defined as no more than 1 loss of balance). Part of my confusion may be due to unfamiliarity with testing in PD. Is it true that individuals will start at Beginner level, perform the exercise, and if completed with no more than 1 loss of balance, they will repeat the exercise at the Easy level (and if completed, move on to the Intermediate level, etc.)? Since not all readers will be familiar with PD, some additional explanation may be helpful (for the control group, it seems clear that they will move from one level to the next level, by the "Progression Principle" column in Table 2).

3. For the sample size calculation, authors should clarify the correlation between measurements (if they in fact used the repeated measures design). I suspect given that authors state the effect size as "d", they are referring to the difference between groups at a particular time point, but this should be clarified.

4. If authors updated their statistical analysis section to say they are using linear mixed effects models, it isn't clear why they are also mentioning repeated measures ANOVA. Authors will be able to get group differences at the different time points within the linear mixed effects model.

5. How will authors report and analyze the adverse events? It seems as though authors only discuss adverse events in the context of IVR? What about adverse events in the control group or adverse events not related to IVR?

**Do you want your identity to be public for this peer review?** For information about this choice, including consent withdrawal, please see our Privacy Policy

Reviewer #2: No

---

## [Author Response · Author response to Decision Letter 2]

27 Jan 2026

We appreciate your review of our manuscript. We also welcome your comments to improve the quality of the manuscript. We have made the requested changes and revised the manuscript accordingly. Attached is a point-by-point response. We hope you find our responses satisfactory.

Reviewers' comments:

Reviewer #2: General Comments

The authors present the protocol for a single-blind two arm randomized controlled trial in individuals with Parkinson's disease, evaluating immersive virtual reality (IVR) based balance training versus standard balance training. The manuscript will be strengthened if the authors consider the following points.

1. In the Eligibility criteria section, authors mention that adherence to the intervention protocol below 90% is a reason for withdrawal. They later say that analyses will be based on the intent-to-treat principle, which includes all individuals who have been randomized (so adherence is not a factor). These two statements are inconsistent and should be clarified.

Response:

Thank you for your comment. After carefully reviewing your observation, we have decided to remove the criterion regarding withdrawal due to adherence below 90%. We understand that this could lead to confusion, as you correctly pointed out that the analysis will be based on the intention-to-treat principle, which includes all individuals who were randomized, regardless of their level of adherence. Therefore, we have revised the Eligibility Criteria section and removed this reference to ensure consistency with the approach of our analysis.

2. Some clarification is still warranted for determining the difficulty level of the exercises, particularly for those in the IVR group. Authors state that individuals will perform all exercises at the highest possible difficulty level (defined as no more than 1 loss of balance). Part of my confusion may be due to unfamiliarity with testing in PD. Is it true that individuals will start at Beginner level, perform the exercise, and if completed with no more than 1 loss of balance, they will repeat the exercise at the Easy level (and if completed, move on to the Intermediate level, etc.)? Since not all readers will be familiar with PD, some additional explanation may be helpful (for the control group, it seems clear that they will move from one level to the next level, by the "Progression Principle" column in Table 2).

Response:

Thank you for this comment. We agree that additional clarification was needed, particularly for readers unfamiliar with exercise progression in Parkinson’s disease. In the IVR group, participants will begin each exercise at the Beginner level and will progress sequentially through increasing difficulty levels if the exercise is completed with no more than one loss of balance. This process will continue until the highest difficulty level that can be safely performed is reached, following the same procedure as for the control group. We have revised the Methods section to explicitly describe this stepwise progression procedure.

3. For the sample size calculation, authors should clarify the correlation between measurements (if they in fact used the repeated measures design). I suspect given that authors state the effect size as "d", they are referring to the difference between groups at a particular time point, but this should be clarified.

Response:

Thank you for this comment. We have clarified that the sample size calculation was based on the expected difference between groups at the post-intervention time point (d = 0.71, α = 0.05, 80% power). Although the study includes repeated measurements at three time points, including follow-up, this calculation provides a conservative estimate that is expected to offer adequate power for comparisons at follow-up. The Methods section has been revised accordingly.

4. If authors updated their statistical analysis section to say they are using linear mixed effects models, it isn't clear why they are also mentioning repeated measures ANOVA. Authors will be able to get group differences at the different time points within the linear mixed effects model.

Thank you for this observation. We have removed the reference to repeated‑measures ANOVA from the Statistical Analysis section. The analysis will rely on linear mixed‑effects models, which allow estimation of group‑by‑time differences across all measurement points. The Methods section has been updated accordingly.

5. How will authors report and analyze the adverse events? It seems as though authors only discuss adverse events in the context of IVR? What about adverse events in the control group or adverse events not related to IVR?

Response:

Thank you for your comment. Adverse events will be systematically monitored in the IVR group, as this intervention involves a novel technology with known adverse effects. In the control group, the exercises are based on validated programs with no previously reported adverse events, as described in the article by Ernst M, Folkerts AK, Gollan R, Lieker E, Caro-Valenzuela J, et al. Physical exercise for people with Parkinson's disease: a systematic review and network meta-analysis. Cochrane Database Syst Rev. 2023 Jan 5;1(1):CD013856. doi: 10.1002/14651858.CD013856.pub2. Updated in: Cochrane Database Syst Rev. 2024 Apr 08;4:CD013856. doi: 10.1002/14651858.CD013856.pub3. Therefore, systematic monitoring is not planned for the control group. However, any unexpected events reported by participants in the control group will also be documented. For participants in both groups, all adverse events will be recorded in detail, including type, severity, duration, and possible relationship to the intervention.

---

## [Decision Letter · Decision Letter 2]

22 Feb 2026

Effects of a level-based immersive virtual reality physical therapy program on static and dynamic balance in Parkinson’s disease: Protocol for a randomized controlled trial

PONE-D-25-61370R2

Dear Dr. Calvo,

We’re pleased to inform you that your manuscript has been judged scientifically suitable for publication and will be formally accepted for publication once it meets all outstanding technical requirements.

Kind regards,

Domna Banakou

Academic Editor

PLOS One

Reviewers' comments:

Reviewer's Responses to Questions

**Comments to the Author**

1. Does the manuscript provide a valid rationale for the proposed study, with clearly identified and justified research questions?

Reviewer #2: Yes

2. Is the protocol technically sound and planned in a manner that will lead to a meaningful outcome and allow testing the stated hypotheses?

Reviewer #2: Yes

3. Is the methodology feasible and described in sufficient detail to allow the work to be replicable?

Reviewer #2: Yes

4. Have the authors described where all data underlying the findings will be made available when the study is complete?

Reviewer #2: Yes

5. Is the manuscript presented in an intelligible fashion and written in standard English?

Reviewer #2: Yes

You may also provide optional suggestions and comments to authors that they might find helpful in planning their study.

Reviewer #2: The authors have addressed all of my earlier comments, and I have no additional concerns about the protocol.

**Do you want your identity to be public for this peer review?** For information about this choice, including consent withdrawal, please see our Privacy Policy

Reviewer #2: No

---

## [Editor Report · Acceptance letter]

PONE-D-25-61370R2

PLOS One

Dear Dr. Calvo,

I'm pleased to inform you that your manuscript has been deemed suitable for publication in PLOS One. Congratulations! Your manuscript is now being handed over to our production team.

Kind regards,

on behalf of

Dr. Domna Banakou

Academic Editor

PLOS One